# Aquatic Exercise on Brain Activity in Type 2 Diabetic: Randomized Clinical Trial

**DOI:** 10.3390/ijerph192214759

**Published:** 2022-11-10

**Authors:** Guilherme Cândido Viana Gonçalves, Adriana Teresa Silva Santos, Ruanito Calixto Júnior, Miqueline Pivoto Faria Dias, Denise Hollanda Iunes, Erika de Cássia Lopes Chaves, Ligia de Sousa Marino, Juliana Bassalobre Carvalho Borges, Andréia Maria Silva Vilela Terra

**Affiliations:** 1Pos-Graduation in Rehabilitation Sciencies, Institute of Motor Science, Federal University of Alfenas, Santa Clara Campus, Alfenas 37133-840, MG, Brazil; 2Human Performance Research Laboratory, Institute of Motor Science, Federal University of Alfenas, Santa Clara Campus, Alfenas 37133-840, MG, Brazil; 3Nursing School, Federal University of Alfenas, Alfenas 37130-001, MG, Brazil

**Keywords:** type 2 diabetes mellitus, electroencephalography, physical training, rehabilitation

## Abstract

Background: A water-based physical exercise program is extremely important for the rehabilitation of type 2 diabetes. Little is known about its action on cerebral electrical activity. Objective: To evaluate the effect of a water-based physical exercise protocol on electroencephalographic activity, blood glucose levels, and functional capacity, as well as their correlation, in type 2 diabetics. Methods: Study design: Randomized Clinical Trial. Forty volunteers were randomized into two groups: control (n = 20) and study (n = 20). A water-based physical exercise program comprising 50 min sessions was conducted three times a week for five weeks. Assessments were performed at the pre- and post-intervention and follow-up phases. The qualitative data were compared using the Mann–Whitney test and Chi-Square. Quantitative data were compared using the Kruskal–Wallis, Independent *t*, and ANOVA mixed tests. The Spearman correlation coefficient was used to correlate the data. Results: The data were similar when comparing the groups. Six-minute walk test data increased in the comparison between times (*p* = 0.01—PrexPos). EEG data decreased in comparison between times (prexfollow-up—*p* < 0.05), except AF3. EEG data decreased in the timexgroup comparison (prexfollow-up and postxfollow-up—*p* < 0.05). Conclusions: The water-based exercise protocol maintained electroencephalographic activity, glucose levels, and functional capacity in people with type 2 diabetes, and there was no relationship between brain electrical activity and capillary blood glucose.

## 1. Introduction

Type 2 diabetes mellitus (DM2) is a heterogeneous polygenic disease strongly associated with genetic susceptibility. The occurrence of DM2 may be related to poor eating habits and physical inactivity, the association of which can lead to obesity, which is the main risk factor for the disease. DM2 is associated with excess weight and other components of the metabolic syndrome in 80–90% of cases. In many countries, it is considered a disease presenting enormous challenges for public health in the 21st century [1].

The worldwide prevalence of diabetics aged 20–79 years was 9.3% (463 million) in 2019, and it is estimated to reach 10.2% (578.4 million) in 2030 [2]. In the Brazilian population aged 55–64 years, the prevalence of diabetes was approximately 17%:18.9% for men, and 16% for women [3]. The impact of this disease can be reversed through extensive interventions, cost-effective early detection, and timely treatment [4].

The scientific literature has advocated physical exercises as a predictor of a healthy life. In this sense, according to the American College of Sports Medicine (ACSM), prescription of physical exercise depends on several factors, namely, frequency, intensity, time, and type, as well as volume and progression (FITT-VP) that are consistent with the disease [5]. Improvements in insulin resistance and functional capacity and reduction in arterial blood pressure levels are directly associated with these factors [6].

According to the guidelines of the American Association of Diabetes Educators for the practice of diabetes self-management education/training, treatments should aim at planning proper nutrition and the correct prescription of regular exercise [7]. However, some people may experience difficulty in performing the exercises, which often impact the joints, causing pain, and many individuals end up abandoning the training protocol [8]. Thus, the prescription of high-impact physical exercises can compromise the joints or the musculoskeletal system [9,10].

Water-based physical exercises can also be prescribed. This approach is influenced by the physical principles of water, which act on all systems of the human body, such as the cardiovascular [11], respiratory, and urinary [12] systems.

A meta-analytic study concluded that methodological factors such as duration and type of exercise and sex can have a direct influence on brain function in elderly individuals [13]. A more recent revisited version of this meta-analysis showed that cognitive function may or may not be influenced by exercise in the population aged ≥80 years, and that information such as methodological factors for prescribing physical exercise needs to be more thorough [14]. A study reported that the brain-derived neurotrophic factor remains constant in type 2 diabetics compared with that in a control group after nine months of aerobic or resistance exercise, or a combination of both [15].

There is a need for further research in this area to engage more comprehensively with this subject, as aquatic exercise has been little explored in the population with diabetes. However, conflicting results have been found in the literature on the methodology of dry-land training in modifying electroencephalographic (EEG) activity in diabetics. With this, we hypothesized that 5 weeks of aquatic training would modify brain electrical activity and we also hypothesized that there would be a correlation with capillary blood glucose.

In this context, this study assessed the effect of a water-based physical exercise protocol on EEG activity, blood glucose levels, and functional capacity, as well as their correlations, in individuals with DM2.

## 2. Materials and Methods

### 2.1. Trial Design 

This was a randomized clinical trial. Randomization (allocation concealment) was individual, and both the outcomes and statistical evaluators were blinded to the study sampling and results. 

This study was conducted at the Therapy Pool, Physical Therapy College, UNIFAL-MG, and at a private swimming academy in the municipality of Alfenas (MG), Brazil, between October 2017 and January 2019. The expected date of first recruitment was 1 August 2017.

Figure 1 shows the drawing and flowchart of the clinical trial participants. Researcher 1 contacted the Basic Health Units in Alfenas, and 2781 participants were initially found to be eligible for the study. Of these, 2545 were excluded for various reasons, such as not meeting the inclusion criteria, being unwilling to participate in the study, presenting contact difficulties (idle phone number, busy phone, no response to calls, or no phone number available), and deceased users. Volunteers of both sexes, age ≥ 18 years, with clinical diagnosis of DM2 for at least five years, having a medical referral, and who agreed to participate after signing the informed consent were included in the study. Exclusion criteria were as follows: patients with infectious or cancerous processes in the active phase, active deep vein thrombosis, dermatitis, or dermatosis in the lower limbs; patients who did not present voluntary movements of the lower limbs and were unable to perform isotonic exercises; patients with wounds or ulcers in the lower limbs, severe trauma, generalized edema secondary to heart or renal failure; and pregnant women. Forty (40) volunteers were randomly allocated into two groups: the control group (CG; n = 20), and the study group (SG; n = 20). Randomization was individual with allocation concealment (generated and code identified by Researcher 2) using randomizer software (www.random.org, accessed on 20 January 2018). The outcome evaluator, Researcher 3 (trained and qualified), was blinded to the sampling. The interventions were conducted by Researcher 4 (skilled and trained). The statistical evaluator was blinded to the data and received the database with no identification codes for the groups.

### 2.2. Pilot Study and Sample Size Calculation

A pilot study was conducted with six individuals for each group. Post-intervention blood glucose level was the variable used to calculate the sample size. Capillary blood glucose levels of 310.50 ± 60.49 (mg/dL) and 181.50 ± 95.48 (mg/dL) were found for the CG and SG, respectively. The result determined 16 participants: eight in the CG and eight in the SG. A significance level of 5% (*α* = 0.05) and a statistical power of 0.95 were adopted. All data were processed using GPower 3.1 software.

### 2.3. Assessment Measures

All evaluations were conducted by an experienced researcher blinded to the results. Protocol assessments were performed at the following times: pre-intervention, post-intervention (after 15 sessions), and follow-up (15 days after the end of intervention).

The following instruments were used to collect the data: anthropometric data, assessment of functional capacity, International Physical Activity Questionnaire (IPAQ), and assessment of EEG activity.

### 2.4. Anthropometric Data

Height and body mass of the participants were measured using a bioimpedance scale (G-Tech^®^—Zhongshan Camry Electronic CO, Zhongshan, China), and the Body Mass Index (BMI) (body mass/height^2^) was calculated. The BMI measurement and classification procedures were conducted in accordance with Flegal [16]. Capillary blood glucose was measured using a G-Tech Free™ glucometer in the morning for fasting.

### 2.5. International Physical Activity Questionnaire (IPAQ)

IPAQ is an instrument used to quantify the levels of physical activity of different specific populations, in addition to making comparisons between different populations at the international level [17]. The application of the IPAQ, which has been validated in 12 countries, provides information on the weekly time spent walking, in moderate- and vigorous-intensity activity, and in sedentary activity. A short form of the questionnaire validated for the Brazilian population was adopted in this study [18].

### 2.6. Assessment of Functional Capacity

Objective evaluation of functional exercise capacity was performed using the 6 min walk test (6MWT) following the American Thoracic Society (ATS) guidelines [19]. Measures of arterial blood pressure (ABP), heart rate (HR), respiratory rate (RR), pulse oximetry (SpO_2_) in percentage (%), and perceived dyspnea and fatigue according to the Borg rating of perceived exertion (RPE) scale [20] were taken at the baseline and the end of the test. The equations defined by Enright and Sherril [20] were applied to calculate the estimated distance. The 6MWT has been validated for the Brazilian population [21]. The evaluation was performed in the morning.

### 2.7. Assessment of Electroencephalographic (EEG) Activity

A commercially available, brain–computer interface headset, the Emotiv EPOC^®^ (EMOTIV Inc. San Francisco, CA, USA) EEG gaming system, was used to assess the EEG activity. This is an easy-to-use, low-cost, non-invasive device consisting of 16 gold-plated contact sensors positioned on specific places of skull that record EEG activity and transmit the data wirelessly to a receiver unit connected to a computer. This instrument has been validated for research [22].

The channel nomenclature was based on the International 10–20 system and included the following 14 electrode positions: AF3, F7, F3, FC5, T7, P7, O1, O2, P8, T8, FC6, F4, F8, and AF4 [23]. Regarding the headset positioning, it had to be placed approximately 6 cm apart between the two frontal electrodes and the eyebrows. The device has an internal frequency of 2048 Hz that is down-sampled to 128 Hz sampling frequency per channel before transmitting the data to the acquisition computer [24]. Only the frontal region of the brain was investigated in this study: AF3—frontal left anterior channel; AF4—frontal right anterior channel; F3—frontal left medial channel; F4—frontal right medial channel; F7—frontal left lateral channel; F8—frontal right lateral channel; FC5—frontal left central channel; and FC6—frontal right central channel. Each electrode presents anatomical correspondence with brain areas: AF3/AF4—superior frontal gyrus, F3/F4—middle frontal gyrus, F7/F8—inferior frontal gyrus, and FC5/FC6—precentral gyrus [25].

Emotiv Xavier Pure^®^ EEG 3.4.3 software (Emotiv, San Francisco, CA, USA) was used to collect the EEG signals.

Electroencephalographic data were collected from participants at rest, seated in a chair with a backrest. The researcher positioned the headset on the participant’s scalp according to the manufacturer’s guidelines. The EEG gaming system was calibrated, and then the data were collected with the participant’s eyes closed for 10 s. This assessment was conducted at the pre- and post-intervention (after 15 sessions of the aquatic physical exercise) and follow-up (15 days after the end of the training protocol) times. The evaluation was performed in the morning. Figure 2 shows the placement of the electrodes.

### 2.8. Processing and Analysis of EEG Data

Electroencephalographic data were analyzed using MATLAB R2017a software and the EEGLab v14.1.1 toolbox. First, in EEGLab, cuts were made, and the central 20 s of the signal were delimited, from which the first five seconds and the last five seconds were excluded. Thus, the analysis was performed after 10 s of EEG signal collection. The presence of noise was verified and eliminated.

Also using the EEGLab, the positions of the channels were established by Independent Component Analysis (ICA). The signals obtained through action potentials registered by each electrode were considered because of the activation of several sources, and they were strongly correlated with each other. ICA is a mathematical computational tool used for separating the recorded multivariate signals referring to the feature, which was considered as approximations of the sources, into components [26]. Subsequently, high- and low-pass filters were applied at the cut-off values at 8 and 30 Hz, respectively.

In MATLAB, the Welch periodogram, which is characterized by reducing the spectrum variance, was applied based on the reduction of spectral resolution. This function transforms the signal over time into a power signal at the displayed frequency [27]. Using MATLAB, a specific routine for the analysis of the signal captured in this study was created based on previous studies of EEG signal analysis [28]. The frequency spectra were analyzed within the bands most relevant to the study. The following frequency bands were considered: Alpha (8–15 Hz), Beta (15–30 Hz), Alpha-High (10–11 Hz), and Alpha-Low (9 Hz) [29]. The frontal, pre-frontal, and primary motor areas were the brain regions assessed. The data were analyzed by power statistics corresponding to the amplitude of the EEG signal.

### 2.9. Training Protocol

#### 2.9.1. Experimental Group (SG)

The training protocol of this study included 60 min sessions conducted three times per week divided into warm-up, aquatic conditioning, and muscle resistance at moderate intensity following a modified BORG RPE scale [20], and relaxation. The program lasted five weeks, and 15 sessions were conducted. The first three sessions were considered an adaptation to the aquatic environment, whereas the remaining 12 sessions were of active intervention. The interventions were conducted at the Therapy Pool, Physical Therapy College, UNIFAL-MG (Pool A) and at a private swimming academy (Pool B) in the municipality of Alfenas. Pool A was 11 m long, 10 m wide, and 1.2–1.8 m deep, whereas Pool B was 10 m long, 5 m wide, and 1.5 m deep; a water temperature of 32 °C was maintained in both pools. The protocol was performed in the morning.

The water-based physical exercise program was composed of four phases: I warm-up (10 min), II aquatic conditioning (30 min), III muscle resistance (10 min), and cool-down/relaxation (10 min) (Figure 3).

PHASE IV—Relaxation was preformed using the Ai Chi method (exercises performed in a symmetric and asymmetric vertical posture associated with diaphragmatic breathing) and global stretches, held for 30 s each.

The training was conducted with individuals immersed up to the xiphoid process. Arterial blood pressure and BORG ratings for the lower limbs and respiration were collected at baseline and the end of each session, with the latter used during aquatic conditioning.

#### 2.9.2. Control Group (CG)

The CG did not receive any intervention during the time the study was conducted, and the participants were treated at the university’s clinic after study completion.

### 2.10. Data Analysis

Data were tabulated in Microsoft Excel^®^ (2016) software. Data were presented as mean, standard deviation, and percentage. The database was analyzed using Statistical Package for the Social Sciences (SPSS 20.0) software. The normality (Shapiro–Wilk) and sphericity (Mauchly) of the data were tested, and some data were corrected with the Greenhouse–Geisser test. Qualitative data (sex, physical activity, smoker, alcoholism, hypertension) were compared using the Chi-square test. Qualitative data IPAQ were compared using the Mann–Whitney test. Quantitative data (age, height, body mass, BMI, and diagnostic time) were compared using the Kruskal–Wallis test and the Independent t-test. The EEG variable data were compared using the Mixed ANOVA test followed by the Sidak test. The Spearman correlation coefficient was used to correlate EEG and 6MWT data. A significance level of 5% (*p* < 0.05) was adopted for all statistical analyses.

## 3. Results

Table 1 shows the baseline characteristics of both groups, where a statistically significant difference can be observed for the IPAQ variable (*p* = 0.04).

Table 2 presents the functional capacity and capillary blood glucose variables. The mixed ANOVA showed that there was a significant increase in the 6MWT variable when comparing the pre-intervention with the post-intervention (F_1.59, 55.71_ = 5.22; *p* = 0.01).

Table 3 shows the EEG activity of the two groups. The Mixed ANOVA showed that the amplitude of measured electric potential decreased for channel AF4 (F_2,70_ = 4.24; *p* = 0.01), channel F7 (F_2,70_ = 5.91; *p* = 0.004), channel F3 (F_2,70_ = 3.51; *p* = 0.03), channel F4 (F_2,70_ = 4.06; *p* = 0.02), channel FC5 (F_2,70_ = 4.82; *p* = 0.01), and channel FC6 (F_2,70_ = 7.32; *p* = 0.001) at pre-intervention with follow-up (Time). Mixed ANOVA showed that the amplitude of measured electric potential decreased of channel AF3 (F_2,70_ = 6.08; *p* = 0.004), channel AF4 (F_2,70_ = 4.24; *p* = 0.01), channel F7 (F_2,70_ = 5.91; *p* = 0.004), channel F3 (F_2,70_ = 3.51; *p* = 0.03), channel F4 (F_2,70_ = 4.06; *p* = 0.02), channel FC5 (F_2,70_ = 4.82; *p* = 0.01), and channel FC6 (F_2,70_ = 7.32; *p* = 0.001) at pre-intervention with follow-up (Groups—Experimental x Time—pre-intervention with follow-up).

Table 4 shows the correlation of FC5 and FC6 channels with capillary glycemia and functional capacity. The Spearman test showed no correlation between the variables.

## 4. Discussion

This manuscript hypothesized that 5 weeks of aquatic training would modify brain electrical activity and correlate with capillary blood glucose. This study demonstrated that five weeks of water-based physical exercise maintained the EEG activity, blood glucose levels, and functional capacity of DM2 in both groups. We found results only in the study group. Brain electrical activity decreased after aquatic training.

These results disagree with those presented in the literature. Some studies have reported that individuals with DM2 benefited from aquatic and dry-land training to reduce blood glucose levels. The studies conducted by Suntraluck et al. [30] and Cugusi et al. [31] found decreased blood glucose levels in fasting DM2 patients after performance of aquatic training. The study of Suntraluck et al. [30] performed land-based and water-based physical exercise three times per week for 12 weeks. The study of Cugusi et al. [31] performed 50 min sessions, three time a week for 12 weeks of an aquatic-based exercise program. Eight weeks of physical exercise decreased blood pressure in DM2; however, glycemic control and OGTT-derived insulin sensitivity needs more investigation [32]. Seven days of aerobic exercise training improves whole-body insulin sensitivity in obese humans with mild DM2 [33]. The effect on insulin sensitivity of a single bout of aerobic exercise lasts 24–72 h, depending on the duration and intensity of the activity [34]. Although the frequency of the protocols used in the aforementioned studies was similar to that of this study, the findings were different. Protocol duration is another factor to be considered, since the literature presents divergent data in this regard. 

Physical activity is the non-pharmacological strategy for glycemic control. Aquatic exercises are a possibility for physical activity, and they have an advantage over dry-land exercises because they have less impact on the joints, thus favoring greater safety and comfort during their execution [35]. It should also be emphasized that aquatic exercises trigger the maximization of metabolic, cardiorespiratory, and neuromuscular effects [9]. In this study, although the EEG activity, blood glucose levels, and functional capacity remained unchanged in the individuals investigated, they reported improvements in day-to-day activities.

Comparison between the groups showed that EEG activity remained constant after the intervention. Physical activity induces changes in EEG activity in type 2 diabetics, and the EEG test has substantially helped us understand the mechanisms involved in the relationship between physical exercise and performance in cognitive tasks [36]. Physically active individuals present greater activation of brain bands compared with sedentary or obese individuals, and the pattern of brain activation is differentiated by physical activity or capacity [37]. However, there are no reports in the literature on whether the training duration impacts EEG activity in healthy individuals or those with diabetes.

Reduced EEG activity (motor area) was verified in the SG, which can be explained by the fact that as such a function is already acquired, EEG activity is reduced, and functional performance is increased [38].

According to [39], acute exercise, presenting the same intensity, causes oxidative damage. Thus, it decreases carbonyl levels and increases the degradation of oxidized proteins. In contrast, continuous training attenuates oxidative damage and the proteins involved in intracellular calcium transport, triggering an adaptive response to physical training in the brain of diabetic rodents.

Physical exercise controls blood glucose levels, insulin resistance, and cardiovascular risk factor [40]. Additionally, high-intensity progressive resistance training conducted three times a week for six months was effective in improving blood glucose levels compared with those of a control group [41]. In this context, according to a study conducted by the American Diabetes Association (ADA) in 2016, the main objective of treating DM2 is to decrease blood glucose levels, which shows the importance of this reduction in association with EEG activity. In our study we did not find a relationship between brain electrical activity with capillary blood glucose. Perhaps the time of intervention could be a factor to consider.

Adaptations to chronic exercise depend on the parameters (intensity, duration, frequency, periodization, and mode) and the characteristics of the individual (presence of disease, fitness, and genetic determinants) [9,30,42,43]. The protocol in our study had light-to-moderate intensity, a frequency of three times, and a duration of 60 min. We did not modify the periodization; this may influence the study. Another parameter to be discussed is the level of immersion. The higher the immersion level, the greater the cardiovascular responses and increases in cerebral blood flow velocities [44]. In the present study, we worked with the level of immersion in the xiphoid process. We believe that this could influence the result.

A relevant point to be considered is the medications in use. Anti-diabetic drugs are used to reduce blood glucose. Other beneficial effects including antioxidant, anti-inflammatory, and anti-apoptotic activity [45]. Diabetic patients may have high blood pressure associated with this disease. Anti-hypertensive drugs have effects on the autonomic nervous system, inhibition of the renin–angiotensin–aldosterone system, and peripheral vasodilation [46]. In the present study, we had volunteers who were associated with diabetes and arterial hypertension. 

The strength of the study was the use of aquatic training for type 2 diabetes, investigating brain electrical activity. The weakness of the study may be related to the length of intervention.

We consider the location of intervention as a limitation of the study. Perhaps training in pools with different depths could interfere with the results.

The intensity, volume, and duration of training can be modified for future studies, and methodologies recommend a larger number of weeks. The implications of the findings indicate that the proposed protocol should be modified with clinical practice in mind.

## 5. Conclusions

A water-based exercise protocol maintained electroencephalographic activity, glucose levels, and functional capacity in people with type 2 diabetes, and there is no relationship between brain electrical activity and capillary blood glucose.

## Figures and Tables

**Figure 1 ijerph-19-14759-f001:**
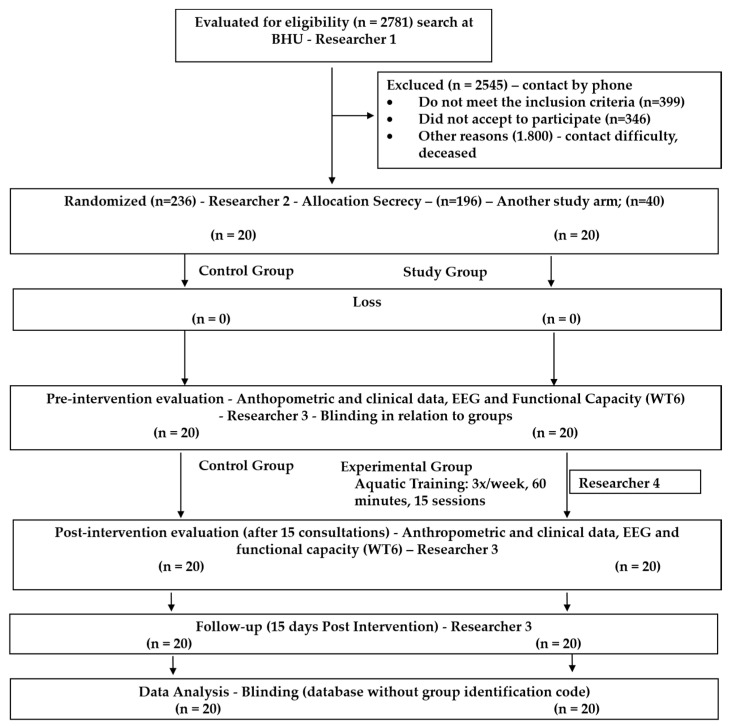
Design and flow chart of clinical trial participants. BHU: Basic Health Unit; EEG: electroencephalogram; WT6: six-minute walk test.

**Figure 2 ijerph-19-14759-f002:**
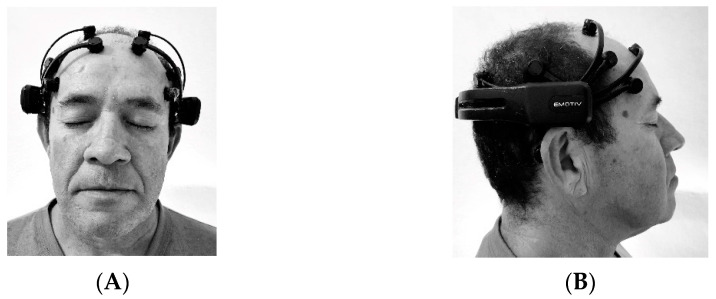
Positions of EEG electrodes: (**A**) front view; (**B**) right side view.

**Figure 3 ijerph-19-14759-f003:**
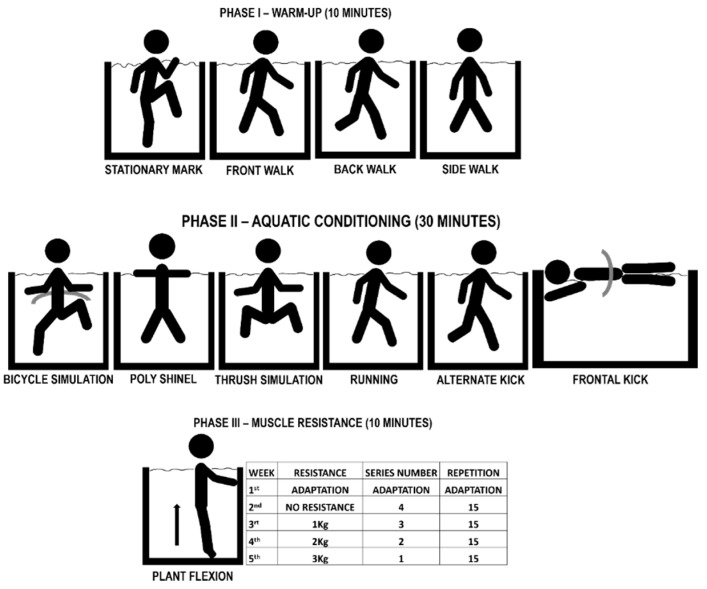
Intervention Group Protocol.

**Table 1 ijerph-19-14759-t001:** Demographic characteristics of participants: control and experimental groups.

Characteristic	Control Group (n = 20)	Experimental Group (n = 20)	*p* Value
Age (yr.), mean (SD), CI 95%	64.20 ± 12.08 58.54 to 69.85	62.22 ± 11.44 57.15 to 67.30	0.59 ^a^
Height (m), mean (SD), CI 95%	1.57 ± 0.09 1.53 to 1.62	1.58 ± 0.09 1.54 to 1.62	0.94 ^b^
Body Mass (Kg), mean (SD), CI 95%	79.77 ± 20.75 70.05 to 89.48	71.25 ± 9.84 66.88 to 75.62	0.10 ^a^
BMI (kg/m^2^), mean (SD), CI 95%	25.10 ± 5.56 22.50 to 27.71	22.47 ± 2.76 21.24 to 23.69	0.05 ^a^
Diagnostic time (yr), mean (SD), CI 95%	11.75 ± 5.96 8.95 to 14.54	10.27 ± 5.24 7.94 to 12.59	0.44 ^b^
Sex, n (%)			
Feminine	12 (60)	13 (65)	0.40 ^c^
Male	8 (40)	7 (35)	
Physical activity, n (%)			
No	11 (55)	10 (50)	0.61 ^c^
Yes	9 (45)	10 (50)	
Smoker, n (%)			
No	17 (85)	16 (80)	0.55 ^c^
Yes	3 (15)	4 (20)	
Alcoholism, n (%)			
No	17 (85)	16 (80)	0.55 ^c^
Yes	3 (15)	4 (20)	
Hypertension, n (%)			
No	6 (30)	8 (40)	0.45 ^c^
Yes	14 (70)	12 (60)	
IPAQ questionnaire, n (%)			
Sedentary	4 (20)	0	0.04 ^a^*
Insufficient active A	9 (45)	8 (40)	
Insufficient active B	5 (25)	10 (50)	
Active	2 (10)	2 (10)	
Very Active			

^a^ Mann–Whitney test; ^b^ Independent *t*-test; ^c^ Chi-square test; * *p* < 0.05; yr—year; m—meter; Kg—kilogram; %—percentage.

**Table 2 ijerph-19-14759-t002:** Mean, SD, and CI (95%) of groups, difference within groups, difference between groups of the variables six-minute walk test, six-minute walk test provided, and capillary blood glucose.

Outcome	Groups								
	Pre-Intervention		Post-Intervention		Follow-Up		*p* Value		
	Con (n = 20)	Exp (n = 20)	Con (n = 20)	Exp (n = 20)	Con (n = 20)	Exp (n = 20)	Time	Groups versus Times	Groups
Six-minute walk test *(*m*)*	434.68 ± 81.02 397.79 to 471.56	440.49 ± 85.68 394.83 to 486.14	459.69 ± 77.54 424.39 to 494.98	460.55 ± 48.90 434.49 to 486.61	450.31 ± 87.08 410.67 to 489.95	468.14 ± 50.05 441.47 to 494.81	0.01 * PrexPos	0.51	0.72
Six-minute walk test provided (m)	454.01 ± 54.29 429.30 to 478.73	491.58 ± 82.09 491.58 to 535.33	453.74 ± 53.73 429.28 to 478.20	491.34 ± 82.45 447.40 to 535.28	454.87 ± 53.97 430.32 to 479.42	491.46 ± 82.34 447.57 to 535.34	0.51	0.57	0.10
Capillary blood glucose (mg/dL)	195.19 ± 79.59 158.95 to 231.42	196.68 ± 77.97 155.13 to 238.23	185.61 ± 70.07 153.72 to 217.51	209.37 ± 80.61 166.42 to 252.32	183.38 ± 58.83 156.60 to 210.16	182.00 ± 72.42 143.41 to 220.58	0.33	0.45	0.70

Mixed ANOVA followed Sidak test; Exp = experimental group; Con = control group; * *p* < 0.05.

**Table 3 ijerph-19-14759-t003:** Mean, SD, and CI (95%) of groups, difference within groups, difference between groups of the variable EEG with eyes closed.

	Groups								
	Pre-Intervention		Post-Intervention		Follow-Up		*p* Value		
Channel	Con (n = 20)	Exp (n = 20)	Con (n = 20)	Exp (n = 20)	Con (n = 20)	Exp (n = 20)	Time	Groups versus Time	Groups
AF3 (µv)	41.29 ± 20.38 32.01 to 50.57	41.50 ± 21.56 30.01 to 53.00	34.35 ± 22.80 23.97 to 44.73	45.64 ± 21.46 34.20 to 57.07	40.39 ± 23.11 29.87 to 50.91	24.21 ± 18.50 14.35 to 34.07	0.050	0.004 * Exp PrexFollow PosxFollow	0.77
AF4 (µv)	43.78 ± 20.97 34.23 to 53.33	44.48 ± 21.85 32.83 to 56.12	35.96 ± 22.77 25.60 to 46.33	45.47 ± 21.05 34.24 to 56.69	39.53 ± 21.36 29.80 to 49.25	25.45 ± 18.84 15.40 to 35.49	0.010 * PrexFollow	0.19	0.80
F7 (µv)	44.20 ± 22.20 34.09 to 54.31	43.50 ± 21.95 31.80 to 55.20	35.04 ± 22.45 24.82 to 45.26	42.52 ± 24.30 29.57 to 55.47	38.71 ± 21.93 28.72 to 48.69	21.02 ± 20.04 10.34 to 31.70	0.004 * PrexFollow	0.010 * Exp PrexFollow PosxFollow	0.52
F8 (µv)	44.78 ± 20.80 35.30 to 54.25	44.32 ± 20.96 33.14 to 55.49	36.51 ± 22.75 26.15 to 46.87	40.43 ± 23.16 28.09 to 52.78	40.41 ± 21.38 30.68 to 50.15	19.87 ± 17.94 10.30 to 29.43	0.004 * PrexFollow	0.010 * Exp PrexFollow PosxFollow	0.27
F3 (µv)	40.66 ± 19.35 31.85 to 49.47	40.33 ± 20.30 29.51 to 51.15	32.10 ± 20.67 22.69 to 41.51	42.46 ± 19.10 32.28 to 52.64	38.58 ± 22.58 28.30 to 48.86	21.76 ± 17.09 12.65 to 30.86	0.030 * PrexFollow	0.004 * Exp PrexFollow PosxFollow	0.64
F4 (µv)	42.75 ± 19.60 33.83 to 51.68	39.69 ± 23.55 27.14 to 52.24	35.09 ± 21.44 25.33 to 44.85	40.90 ± 21.58 29.39 to 52.40	39.51 ± 21.30 28.81 to 48.21	21.10 ± 19.28 10.82 to 31.37	0.020 * PrexFollow	0.020 * Exp PrexFollow PosxFollow	0.34
FC5 (µv)	39.93 ± 20.61 30.55 to 49.31	42.58 ± 24.01 19.79 to 55.37	31.514 ± 20.81 22.04 to 40.98	40.36 ± 22.96 28.12 to 52.60	38.80 ± 22.96 28.35 to 49.26	18.10 ± 20.20 7.34 to 28.87	0.010 * PrexFollow	0.002 * Exp PrexFollow PosxFollow	0.57
FC6 (µv)	43.15 ± 21.98 33.14 to 53.16	42.80 ± 22.71 30.70 to 54.90	34.22 ± 21.53 24.42 to 44.02	42.44 ± 24.49 29.38 to 55.49	36.29 ± 21.37 26.56 to 50.86	18.30 ± 18.95 8.20 to 28.40	0.001 * PrexFollow	0.009 * Exp PrexFollow PosxFollow	0.53

Mixed ANOVA followed by Sidak test; AF3—frontal left anterior channel; AF4—frontal right anterior channel; F3—frontal left medial channel; F4—frontal right medial channel; F7—frontal left lateral channel; F8—frontal right lateral channel; FC5—frontal left central channel; FC6—frontal right central channel; Exp = experimental group; Con = control group, * *p* < 0.05.

**Table 4 ijerph-19-14759-t004:** Correlation of FC5 and FC6 channels with capillary glycemia and functional capacity.

Outcome		Group			
		Con	Exp	Con	Exp
		Post	Follow-Up	Post	Follow-Up
Channel EEG x Functional Capacity					
FC5 x 6MWT	r *p*	−0.05 0.83	0.2 0.41	−0.20 0.38	0.27 0.25
FC6 x 6MWT	r *p*	0.19 0.40	0.2 0.41	−0.01 0.96	0.13 0.57
Channel EEG x Glycemia Capillary					
FC5 x Glycemia Capillary	r *p*	0.25 0.28	0.28 0.24	0.04 0.84	0.18 0.44
FC6 x Glycemia Capillary	r *p*	0.33 0.14	0.21 0.37	−0.09 0.68	−0.04 0.85

Spearman test; 6MWT = 6 min walk test; FC5—frontal left central channel; FC6—frontal right central channel; EEG—brain electrical activity; Exp = experimental group; Con = control group.

## Data Availability

Study data and the statistical analysis plan can be shared upon request to andreia.silva@unifal-mg.edu.br.

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
