# Peer review of "Aquatic Exercise on Brain Activity in Type 2 Diabetic: Randomized Clinical Trial"

_ijerph, 2022, doi:10.3390/ijerph192214759_

Round 1
Reviewer 1 Report
The manuscript by Gonçalves et. al identifies the effect of water based physical exercise on EEG activity and blood glucose levels in type 2 Diabetic patients. The study is well explained, but authors need to take care of a few comments.
1. Authors need to mention what criteria was used to confirm that the patients were type 2 diabetic.
2. Were the included individuals on any medication during the course of the study? If so, the effects of the medication should be explained.
3. In method section authors mention n=20 for control and experimental group whereas the tables 2 and 3 state otherwise. Explain.
4. Table 4 labelling “Con, Exp. Pos. follow up” needs clarification/correction.
5. Regarding EEG activity in table 3, as the control group was not involved in the physical aquatic exercise, why is there a reduction in mean EEG activity in the control group in pre intervention vs. post intervention? Explain.
6. In the discussion section, authors need to discuss more as what could be the reason behind observed results.
Author Response
Dear reviewer, thank you for your comments. I'm sure it contributed a lot to the work
Point 1. Authors need to mention what criteria was used to confirm that the patients were type 2 diabetic.
Answer 1. line 90 and 91 - clinical diagnosis of DM2 for at least five years, having medical referral
Point 2. were the included individuals on any medication during the course of the study? if so, the effects of the medication should be explained.
Answer 2. We add ourselves to the discussion.
Point 3. in method section authors mention n=20 for control and experimental group whereas the tables 2 and 3 state otherwise. explain.
Answer 3 - We made the correction, there was an error in inserting the data in the table
Point 4. table 4 labelling “con, exp. pos. follow up” needs clarification/correction.
Answer 4. we correct the table 4 labelling
Point 5. Regarding EEG activity in table 3, as the control group was not involved in the physical aquatic exercise, why is there a reduction in mean EEG activity in the control group in pre intervention vs. post intervention? explain.
Answer 5. dear reviewer, checking table 3, there was no difference in the control group between times (pre-intervention with post-intervention).
Point 6. In the discussion section, authors need to discuss more as what could be the reason behind observed results.
Answer 6. We add some reflections to be considered by the result obtained.
Adaptations to chronic exercise depend the parameters (intensity, duration, frequency, periodization and mode) and the characteristics of the individual (presence of disease, fitness, and genetic determinants) [9,30,42,43]. The protocol in our study has light to moderate intensity, frequency three times, duration of 60-min. We did not modify the periodization; this may influence the study. Another parameter to be discussed is the level of immersion. The higher the immersion level the greater the cardiovascular responses and increase in cerebral blood flow velocities [44]. In the present study, we worked with the level of immersion in the xiphoid process. We believe that this could influence the result.

Reviewer 2 Report
The Authors undertook a research in a very important field aimed at reducing diabetic complications in people with type 2 diabetes. Physical activity is well-known health benefit, including diabetes. Authors studied the influence of exercise in water not only on fasting glycemia, but also on many other parameters including EEG. Here are my comments:
Mayor
- "The program lasted five weeks and 15 sessions were conducted". What was the rationale for the study duration? Could such a period be effective? Maybe Authors could give some examples that such short period was effective in earlier studies in persons with type 2 diabetes. By contrary there is an example (Ref. 38) that 6 months of exercises was effective. Therefore, the Authors’ finding is that 5 week period is too short to give a significant change.
- Pilot study and sample size: "a statistical power of 0.95 were adopted" - Usually the power of 0.80 is used. Why Authors used 0.95?
- A figure showing the positions of EEG electrodes should be provided.
Minor
- Pilot study and sample size: The BG levels are given for CG and SG, but the units are omitted.
- What units are when EEG electrodes are mentioned?
- "gold-plated contact sensors positioned on specific areas of the brain" - rather surface of skull, not brain, isn't it?
- Table 1: Authors use Kg and kg for kilogram.
- Table 1: Authors write in column 1 "mean (SD)" in description of some parameters, but in column 2 the data is given as mean ± SD. Correct this, please.
- English language proofreading is necessary to avoid, for instance, such expressions as in Conclusions: "... we too found no relationship ...."
- Results/First sentence:"Table 1 shows the baseline characteristics of both groups, where a statistically significant difference can be observed for IPAQ variable (p=0.01)." Instead in the Table there is 0.04 in the row showing the IPAQ. Which is true?
- Improve the writing style, for example: "The studies conducted by [30,31]" The studies were not conducted by reference numbers. Another example: "Mixed ANOVA showed significant reduction of channel AF4" - What exactly reduction of channel means? Is this means that the amplitude of measured electric potential decreased?
Author Response
Dear reviewer, thank you for your comments. I'm sure it contributed a lot to the work
MAYOR
Point 1. "The program lasted five weeks and 15 sessions were conducted". What was the rationale for the study duration? Could such a period be effective? Maybe Authors could give some examples that such short period was effective in earlier studies in persons with type 2 diabetes. By contrary there is an example (Ref. 38) that 6 months of exercises was effective. Therefore, the Authors’ finding is that 5 week period is too short to give a significant change.
Answer 1 - We hypothesized 5 weeks would be effective for parameter changes. We found data inconsistency in the literature. 6 months, 12 weeks, 8 weeks, 7 days and acute effect of exercise. We have inserted studies line 295 to 301 studies to justify.
The study of Suntraluck et al. [30] performed Land-Based and Water-Based physical exercise three times per week for 12 weeks. The study of Cugusi et al [31] perform 50-minute session, three time a week for 12 weeks of aquatic-based exercise program. Eight weeks physical exercise decrease in blood pressure in DM2, however, glycemic control and OGTT-derived insulin sensitivity need more investigation [32]. Seven days aerobic exercise training improves whole-body insulin sensitivity in obese humans with mild DM2 [33]. The effect on insulin sensitivity of a single bout of aerobic exercise lasts 24–72 h, depending on the duration and intensity of the activity [34].
Point 2. Pilot study and sample size: "a statistical power of 0.95 were adopted" - Usually the power of 0.80 is used. Why Authors used 0.95?
Answer 2. We determined a power of 0.95 because the larger the sample, we would have greater precision in the confidence interval and lower probability of type II error.
Point 3. A figure showing the positions of EEG electrodes should be provided.
Answer 3. We insert in the text a figure
MINOR
Point 1. Pilot study and sample size: The BG levels are given for CG and SG, but the units are omitted.
Answer 1. The units were placed
Point 2. What units are when EEG electrodes are mentioned?
Answer 2. The units were placed in table 3
Point 3. "gold-plated contact sensors positioned on specific areas of the brain" - rather surface of skull, not brain, isn't it?
Answer 2. Fixed on line 154
Point 4. Table 1: Authors use Kg and kg for kilogram.
Anwer 4. Yes. we describe the units in the legend of table 1 - yr – year; m – meter; Kg – kilogram; % - percentage.
Point 5. Table 1: Authors write in column 1 "mean (SD)" in description of some parameters, but in column 2 the data is given as mean ± SD. Correct this, please.
Anwer 5. We made the corrections.
Point 6. English language proofreading is necessary to avoid, for instance, such expressions as in Conclusions: "... we too found no relationship..."
Anwer 6. We made the corrections.
Point 7. Results/First sentence:"Table 1 shows the baseline characteristics of both groups, where a statistically significant difference can be observed for IPAQ variable (p=0.01)." Instead in the Table there is 0.04 in the row showing the IPAQ. Which is true?
Answer 7. We made the corrections. True – 0.04
Point 8. Improve the writing style, for example: "The studies conducted by [30,31]" The studies were not conducted by reference numbers. Another example: "Mixed ANOVA showed significant reduction of channel AF4" - What exactly reduction of channel means? Is this means that the amplitude of measured electric potential decreased?
Answer 8. We made the corrections.
